# Dense and Indiscernible Object Counting in Agricultural Scenes

## Abstract

Object counting in computer vision has traditionally focused on clearly visible objects. Many real-world applications, such as crop yield estimation and fruit harvest planning in agricultural, involve dense and indiscernible object counting (DIOC). These objects are characterized by their small size, dense distribution, and visual ambiguity with surroundings, which makes traditional counting methods impractical. To facilitate research in this crucial yet unexplored challenge, we introduce DIOCblueberry, a specialized dataset that significantly surpasses existing datasets in complexity. Compared to FSC147, the most comprehensive general counting dataset, DIOCblueberry contains 1.9 times more objects per image with an average of 108 instances, while its box pixel ratio of 2.38‰ is 7.9 times smaller. State-of-the-art counting methods struggle significantly on such challenging scenarios, with high counting errors. To address these challenges, we propose MaskCount, a two-stage multi-modal method. The first stage segments objects from complex backgrounds using multi-modal features, while the second stage enhances feature robustness through contrastive loss. We also design an edge-aware patch cropping mechanism for accurate counting of dense and small objects. Extensive experiments demonstrate that MaskCount achieves substantial improvements over previous state-of-the-art methods, reducing MAE and RMSE by 25.13% and 35.17% respectively on DIOCblueberry. We will release our data, models, and code to the public.

## 1 Introduction

Object counting, which aims to estimate the number of instances in an image, has been a fundamental computer vision task. It serves both as a standalone application and an auxiliary component in complex vision systems Sun et al. (2023). As a standalone task, object counting has demonstrated its significance in diverse domains, including surveillance Wang & Wang (2011), crowd analysis Chan et al. (2008), wildlife monitoring Norouzzadeh et al. (2018), dietary assessment Nguyen et al. (2022), and biomedical analysis Alam & Islam (2019). As an auxiliary component, it enhances the performance of instance segmentation Cholakkal et al. (2019), action localization Narayan et al. (2019), and pedestrian detection Xie et al. (2020). Recent advances in large-scale datasets Ranjan et al. (2021); Hsieh et al. (2017); Bargoti & Underwood (2017); Wu et al. (2023) and deep learning techniques Liu et al. (2022); ukić et al. (2023); Jiang et al. (2023); You et al. (2023); Wang et al. (2024); Xu et al. (2024) have significantly improved object counting performance.

Current object counting mainly focuses on general counting with clear visible objects, as illustrated at the *top* and *middle* of Figure 1. These general counting scenarios typically involve objects that are easily distinguishable, with average box pixel ratios ranging from 5‰ to 25‰ and moderate average object counts between 10 to 60 instances per image. However, many real-world applications, especially in agricultural scenes, involve dense and indiscernible object counting (DIOC), where objects are characterized by significantly smaller sizes with average box pixel ratio less than 2.4‰. The number of objects in DIOC scenarios often exceeds hundreds or even thousands per image, and these objects exhibit strong visual ambiguity with surroundings, as shown at the *bottom* of Figure 1. In agriculture, accurate counting is essential for crop yield estimation, harvest planning, and resource allocation Farjon et al. (2020); Linker (2017); Xiong et al. (2019). For instance, precise fruit counting enables farmers to optimize labor allocation and forecast market supply, while crop quantity monitoring facilitates data-driven deci-

sions in irrigation and fertilization Eli-Chukwu (2019); Elavarasan & Vincent (2020); Kamilaris & Prenafeta-Boldú (2018); Van Klompenburg et al. (2020). These characteristics pose unprecedented challenges to traditional counting methods, which are typically designed for clearly visible objects with distinct boundaries and sufficient inter-object spacing. Given these significant implications for both production efficiency and resource utilization in smart agriculture, we propose the DIOC task as a new research direction in computer vision, specifically addressing the challenges of counting objects that are small, dense, and visually ambiguous with their surroundings.

To facilitate research on DIOC, we introduce DIOCblueberry, a specialized dataset focusing on blueberry counting. We choose blueberries as our initial research target because they represent one of the most challenging scenarios in DIOC: their small size makes them difficult to detect, their clustered growth pattern leads to extreme density, and their color changes during ripening creates significant visual ambiguity with leaves and branches. In future work, we plan to expand our dataset to other challenging DIOC scenarios, such as rice panicles, coffee beans, and grape clusters, which share similar characteristics. Through careful collection and rigorous quality control, we have assembled 6,265 images with an average resolution of 1840×1492 pixels. The dataset contains 679,030 meticulously annotated center points, requiring approximately 1,700 human hours for collection, cleaning, and annotation. Our DIOCblueberry demonstrates remarkable complexity with an average of 108 instances per image and an extremely small box pixel ratio of 2.38‰. As illustrated in Figure 1, these characteristics make

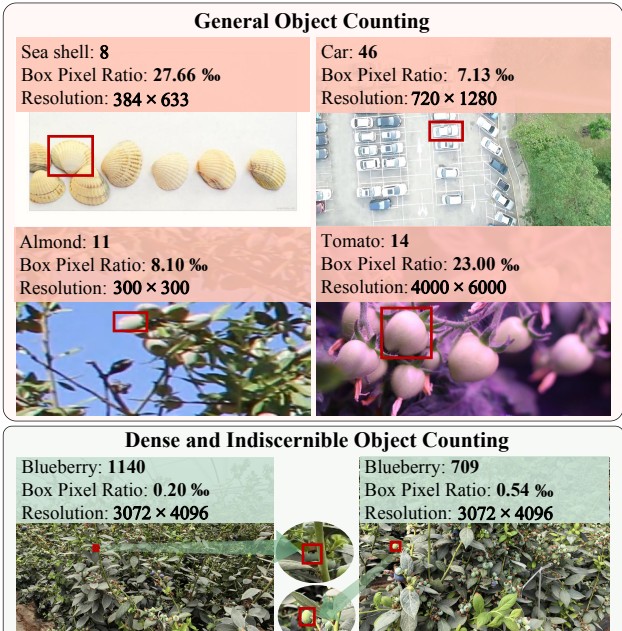

Figure 1: Examples of general counting datasets and our DIOCBlueberry. Box Pixel Ratio refers to the average pixel area of the bounding boxes relative to the total image area. *Top left*: FSC147, *top right*: CARPK, *middle left*: almond dataset, *middle right*: tomato detection dataset, *bottom*: our DIOCBlueberry.

our dataset particularly challenging, as it combines the difficulties of minimal object size, extreme object density, and substantial visual ambiguity with the surrounding environment.

We evaluate six state-of-the-art counting methods on DIOCblueberry. All methods show poor performance with Mean Absolute Error (MAE) exceeding 50, and some methods even produce errors close to 500. Figure 2 illustrates these significant limitations in their counting results. To address these substantial performance gaps, we propose MaskCount, a two-stage multi-modal method that leverages visual specialists and large language models. In the first stage, we utilize CLIP for segmentation to reduce background interference, generating a background mask that simplifies the image for subsequent counting. The second stage enhances feature robustness through contrastive loss, maximizing the separation between objects and background features. Given that real-world applications typically involve high-resolution images, traditional cropping methods often lead to counting inaccuracies at patch edges due to insufficient context information Wang et al. (2021). Therefore, we design an edge-aware patch cropping mechanism that generates overlapping patches and stitches only valid regions to produce the final density map.

We conduct extensive experiments to validate the effectiveness of MaskCount. Our experimental results demonstrate that MaskCount significantly outperforms six state-of-the-art counting methods on DIOCblueberry, reducing MAE and RMSE by 25.13% and 35.17% respectively. Beyond DIOC scenarios, MaskCount also achieves superior performance on general counting datasets in agriculture,

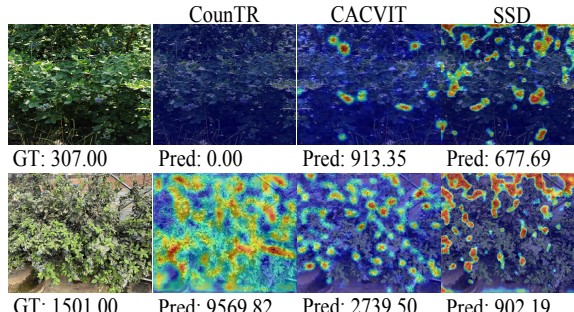

CounTR  CACVIT  SSD

GT: 307.00  Pred: 0.00  Pred: 913.35  Pred: 677.69

GT: 1501.00  Pred: 9569.82  Pred: 2739.50  Pred: 902.19

Figure 2: Examples of counting results from several state-of-the-art counting methods. *First column*: original images, *second column*: CounTR, *third column*: CACVIT, *last column*: SSD.

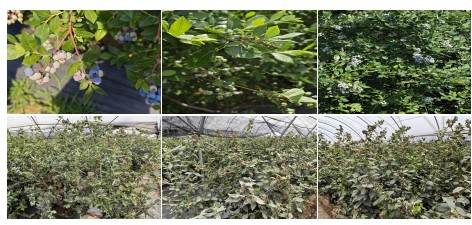

Figure 3: Example images from the proposed DIOCBlueberry. *Top left*: less indiscernible and less tiny sample, *top middle*: indiscernible and less tiny sample, *top right*: less indiscernible and tiny sample, *bottom*: indiscernible and tiny samples (typical samples).

including the almond dataset and tomato detection dataset. Through comprehensive ablation studies, we verify the effectiveness of each key component: the CLIP-based mask generation reduces MAE by 6.33%, the edge-aware patch cropping mechanism further decreases MAE by 14.93%, and our contrastive loss strategy contributes to achieving optimal MAE performance of 38.34.

In summary, the main contributions of this work are as follows:

- We propose the dense and indiscernible object counting (DIOC) task and introduce DIOCblueberry, a specialized dataset for studying dense distribution and visual ambiguity in agricultural scenes.

- We propose MaskCount, a two-stage multi-modal method that segments objects from complex backgrounds using CLIP in the first stage and enhances feature robustness through contrastive loss in the second stage, along with an edge-aware patch cropping mechanism for accurate counting.

- We demonstrate that MaskCount significantly outperforms six state-of-the-art methods on DIOCblueberry, reducing MAE and RMSE by 25.13% and 35.17% respectively, while also achieving superior performance on other agricultural counting datasets.

## 2 DIOCBLUEBERRY

### 2.1 IMAGE COLLECTION

We collected the images for this study using Xiaomi 13 Ultra and Huawei Mate 60 smartphones. The images were gathered from two regions in China: Yunnan Province and Lianyungang City, Jiangsu Province, both known for blueberry cultivation. The images originates from extensive fieldwork on two large farms (500 acres each), capturing genuine agricultural scenarios essential for yield prediction. Unlike many domains, such images cannot be easily scraped; this real-world grounding is a crucial, difficult aspect of valuable agricultural datasets.

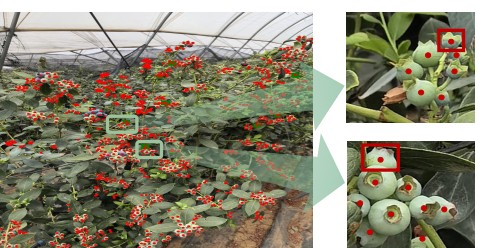

Figure 4: Annotation example, with point and box annotations displayed in red.

The images were captured under various lighting conditions, covering the full range of blueberry growth stages, from unripe to fully ripe. This ensures a comprehensive dataset for the DIOC task. Eight professional annotators initially collected a large number of images, after which they carefully reviewed the dataset and removed those that were unsatisfactory or redundant. This process took a total of 300 human hours. The final dataset consists of 6,265 images, some of which are shown in Figure 3. Additional examples of DIOCblueberry images can be found in Appendix A.1.1.

## 2.2 Image Annotation and Analysis

The LabelMe tool was used for annotation. For each image, 3 objects were arbitrarily selected as exemplars, and axis-aligned bounding boxes were drawn for these instances. The remaining objects were annotated with point annotations. In cases of occlusion, an instance was counted and annotated only if less than 90% of it was occluded. While crowd scenes already involve high object density, our scenario poses even greater complexity owing to the severe indistinguishability of objects. This results in substantially more difficult annotation, where each object demands extensive time and multiple rounds of strict validation to achieve accuracy. Figure 4 illustrates an example of image annotation.

The annotation process was divided into three stages. First, eight professional annotators were trained to familiarize themselves with their tasks. They were trained on knowledge about blueberry growth and well-annotated samples. They were then asked to annotate 15 images each. The annotations were checked and evaluated. Once the annotator had passed the evaluation, they were allowed to proceed to the next stage. Secondly, the images were distributed to eight annotators, with each annotator being responsible for a portion of the dataset. Annotators were required to discuss confusing cases and reach a consensus. Finally, the annotations were reviewed and refined in two rounds. The second stage required 700 human hours, while the third stage required 350 human hours per round. The total cost of the annotation process amounted to 1,400 human hours.

The dataset consists of 6,265 images, with an average height of 1840 pixels and an average width of 1492 pixels. The dataset contains a total of 679,030 objects, with the maximum number of objects in a single image being 1,980.

Table 1: Statistics for existing counting datasets

| Dataset | Images | Avg. Res. | Count Statistics | | | |
|---------|--------|-----------|-------|-----|-----|-----------------|
| | | | Total | Ave | Max | Avg. Box($\%_{oo}$) |
| FSC147 | 6,146 | $384 \times 523$ | 344,150 | 56 | 3,701 | 18.76 |
| CARPK | 1,448 | $720 \times 1280$ | 89,774 | 62 | 188 | 4.59 |
| almond dataset | 620 | $300 \times 300$ | 4,777 | 8 | 37 | 7.24 |
| tomato detection dataset | 520 | $3406 \times 4726$ | 9,112 | 18 | 94 | 14.17 |
| DIOCblueberry (our) | **6,265** | $1840 \times 1492$ | **679,030** | 108 | 1,980 | **2.38** |

Our training set consists of 3,759 images. A total of 35 high-resolution images, averaging $3391 \times 3771$ pixels, were carefully selected by professional annotators to form a test set covering diverse and challenging scenarios, as shown at the *bottom* of Figure 1. These images exhibit high object density and varied spatial distributions, with an average of 297 objects per image and an average box pixel ratio of $0.82\%_{oo}$. Notably, the test set includes the image with the highest object count in the entire dataset (see Table 1), further highlighting its difficulty.

The majority of images in general counting datasets contain fewer than 100 objects. In contrast, a significant proportion of images in our DIOCblueberry dataset contain more than 100 objects, and some even more than 1000. The proportion of images within each object count range across different datasets is provided in Appendix A.1.1.

We compare DIOCblueberry with four general counting datasets. FSC147 Ranjan et al. (2021) is specifically designed for few-shot counting, containing 147 object categories and 6,135 images. CARPK Hsieh et al. (2017) focuses on vehicle counting in parking lots, with rectangular bounding boxes provided for each vehicle. The ACFR Orchard Fruit Dataset Bargoti & Underwood (2017), provided by the agriculture team at the Australian Centre for Field Robotics, The University of Sydney, Australia. It includes apples, mangoes, and almonds, with almond dataset being used for comparison. Tomato detection dataset Wu et al. (2023) contains images of miniature tomatoes, captured under complex lighting conditions in a plant factory. A visual comparison between DIOCblueberry and other counting datasets is provided in Appendix A.1.2.

Table 1 presents a comparison between our DIOCblueberry dataset and four general counting datasets. DIOCblueberry contains a large number of average object annotations, with the average box pixel ratio being much lower than in general counting datasets. This suggests that the objects

are densely distributed and small in size. Additionally, DIOCblueberry exhibits visual ambiguity with the background, which makes the objects harder to distinguish.

In summary, we propose the first specialized dataset for counting dense and indiscernible objects, which is more complex than any existing general counting dataset. Consequently, the substantial human effort dedicated to the challenging on-farm data acquisition and meticulous annotation not only underscores the dataset's complexity, but also matches the scale of labor typically associated with larger benchmarks—highlighting the intrinsic difficulty of curating high-quality datasets tailored for the agricultural DIOC task.

## 3 PROPOSED METHOD

We propose MaskCount, a two-stage multi-modal counting method. As shown in Figure 5, the first stage segments objects from backgrounds and generates a background mask to simplify the image for counting. In the second stage, we introduce a contrastive loss to maximize the separation between objects and backgrounds. Additionally, we design an edge-aware patch cropping mechanism that generates overlapping patches to further improve counting accuracy. In the following sections, we detail the architectures of *Crop* and *Stitch* (edge-aware patch cropping mechanism), as well as *Stage 1* (CLIP-based mask generation) and *Stage 2* (estimating density maps with masked images).

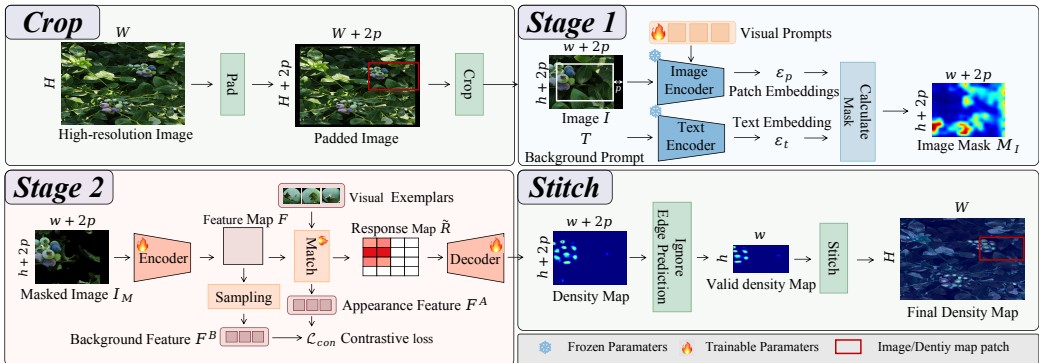

Figure 5: Overview of MaskCount. A high-resolution image is cropped into overlapping patches, which serve as the input. The input image $\mathbf{I}$ is matched with the background prompt $\mathbf{T}$ at the patch level, activating the successfully matched patches to generate the image mask $\mathbf{M_I}$. The masked image is encoded into features $\mathbf{F}$. $\mathbf{F}$ is matched with visual exemplars to generate the response map $\tilde{\mathbf{R}}$, while the appearance features $\mathbf{F^A}$ of the exemplars are encoded simultaneously. $\tilde{\mathbf{R}}$ is decoded to generate the density map. The valid regions of the density maps are stitched together to produce the final density map. During training, the background feature $\mathbf{F^B}$ is extracted from $\mathbf{F}$, sampled, and used to compute a contrastive loss with $\mathbf{F^A}$, aiming to maximize the distance between objects and the background.

### 3.1 *Crop* AND *Stitch*: EDGE-AWARE PATCH CROPPING MECHANISM

General cropping methods result in low counting accuracy at the edges of cropped image patches and noticeable stitching artifacts in the final density map. This is due to the lack of contextual information in edge regions. To address this, we propose an edge-aware patch cropping mechanism that uses a sliding window to generate overlapping image patches. During stitching, only the effective parts of the predicted density maps from image patches are used.

As shown in Figure 5, the high-resolution image size is $(H, W)$. To ensure completeness of the final predicted density map, we add black padding of size $p$ around the high-resolution image, resulting in a padded image of size $(H + 2p, W + 2p)$. The padded image is cropped into $m \times n$ patches using a sliding window of size $(h + 2p, w + 2p)$, where $H = h \times m$ and $W = w \times n$. The horizontal stride of the window is $w$, and the vertical stride is $h$. In other words, overlapping image patches of size $(h + 2p, w + 2p)$ are obtained, with the central region $(h, w)$ being effective. Predictions in the edge regions, with a width of $p$, are discarded, eliminating edge influence and ensuring counting accuracy.

## 3.2 *Stage 1*: CLIP-BASED MASK GENERATION

The primary challenges in DIOC arise from objects being small in size, densely distributed, and exhibiting visual ambiguity with their surroundings. Many real-world applications, such as crop yield estimation, face challenges from highly cluttered backgrounds. These backgrounds include non-object instances like leaves and bushes. Objects with similar colors to backgrounds also contribute to visual ambiguity. Additionally, the small size and dense distribution of the objects complicate subsequent counting tasks. To mitigate these issues, we segment objects from complex backgrounds, thereby reducing the impact of background complexity and easing subsequent counting tasks.

For each high-resolution image, we utilize the large vision-language model Qwen2.5-VL-72B to generate a list of background descriptive words. Specifically, we calculate the similarity between the image and each candidate text. The candidate texts are sorted by similarity, and the top-ranked text is selected as the final background prompt.

As shown in Figure 5, the input image $\mathbf{I}$ and background prompt $\mathbf{T}$ are encoded into patch embeddings $\varepsilon_p$ and text embedding $\varepsilon_t$, respectively. The image and text belong to different modalities, requiring alignment to establish a relationship between them. The cosine similarity map between patch embeddings and the text embedding is calculated to measure the matching degree between image patches and text. Bilinear interpolation is applied to the cosine similarity map, resizing the result to match the input image size, yielding the image mask $\mathbf{M_I}$.

We use InfoNCE loss function He et al. (2020). Minimizing the InfoNCE loss brings positive patch embeddings closer to the text embedding while pushing negative patch embeddings further apart.

$$\mathcal{L}_{stage1} = -\log \frac{\sum_{i=0}^{m} \exp(sim(\varepsilon_{p_+}^i, \varepsilon_t)/\tau)}{\sum_{j=0}^{m+n} \exp(sim(\varepsilon_p^j, \varepsilon_t)/\tau)} \tag{1}$$

where $\{\varepsilon_{p_+}^0, \varepsilon_{p_+}^1, \varepsilon_{p_+}^2, \dots\}$ is the set of positive patch embeddings, $m$ is the number of positive patch embeddings, $\{\varepsilon_p^0, \varepsilon_p^1, \varepsilon_p^2, \dots\}$ is the set of all patch embeddings, $n$ is the number of negative patch embeddings, $\varepsilon_t$ is the text embedding, $sim(\varepsilon_p^i, \varepsilon_t)$ denotes the computation of the cosine similarity matrix between $\varepsilon_p^i$ and $\varepsilon_t$, and $\tau$ is a temperature hyper-parameter per.

## 3.3 *Stage 2*: ESTIMATING DENSITY MAP WITH MASKED IMAGE

As shown in Figure 5, the masked image is encoded into a feature map. The appearance feature $\mathbf{F}^A$ of visual exemplars is extracted using RoI Pooling. The shape feature $\mathbf{F}^S$ of visual exemplars is extracted using MLP. The feature map, appearance feature, and shape feature undergo cross-attention blocks to extract exemplar prototypes. The process of the cross-attention blocks is described as follows:

$$\mathbf{Q}_\ell' = \mathrm{MHA}\left(\mathrm{LN}\left(\mathbf{Q}_{\ell-1}\right), \mathbf{F}^A, \mathbf{F}^A\right) + \mathbf{Q}_{\ell-1} \tag{2}$$

$$\mathbf{Q}_\ell'' = \mathrm{MHA}\left(\mathrm{LN}\left(\mathbf{Q}_\ell'\right), \mathbf{F}, \mathbf{F}\right) + \mathbf{Q}_\ell' \tag{3}$$

$$\mathbf{Q}_\ell = \mathrm{FFN}\left(\mathrm{LN}\left(\mathbf{Q}_\ell''\right)\right) + \mathbf{Q}_\ell'' \tag{4}$$

where the inputs at $\ell = 0$ are initialized by the shape feature $\mathbf{Q}_0 = \mathbf{F}^S$, MHA is the standard multi-head attention, LN is layer normalization and FFN is a small feed-forward network. Such a cross-attention blocks structure we used three to get the exemplar prototypes.

The response map $\tilde{\mathbf{R}}$ is obtained by matching the exemplar prototypes with the feature map. Then, the density map is derived by decoding the response map.

To further increase the separation between the objects and the backgrounds, we apply a contrastive loss. Minimizing the contrastive loss between the background feature $\mathbf{F}^B$ and the appearance feature $\mathbf{F}^A$ increases the separation between objects and backgrounds. The background feature in the feature map is extracted using the image mask. $\mathbf{F}^B$ is sampled uniformly to match the number of appearance features $\mathbf{F}^A$, preserving spatial distribution characteristics. The contrastive loss is shown as follows:

$$\mathcal{L}_{con} = -\frac{1}{N} \sum_{i,j} \log\left(1 - \sigma(\frac{\mathbf{F}_i^A(\mathbf{F}_j^B)^T}{\tau}) + \epsilon\right) \tag{5}$$

where $\mathbf{F}_i^A(\mathbf{F}_j^B)^T$ is the dot product (similarity) of appearance feature and background feature. $\tau$ is a temperature hyper-parameter per. $\sigma(\cdot)$ is a sigmoid function. $\epsilon = 10^{-6}$ is a numerical stability term. $N$ is the number of all possible appearance-background pairs.

We use $L_{MSE}$ loss function, which measures the $l_2$ difference between the predicted and ground truth density maps. Each cross-attention block generates a density map. The density map produced by the final cross-attention block serves as the output of our model.

The final loss is a weighted sum of the two components, with the contrastive loss weight $\lambda_{con}$ controlling their relative contributions. The final loss function is defined as follows:

$$\mathcal{L}_{stage2} = \mathcal{L}_{MSE} + \lambda_{con}\mathcal{L}_{con} \qquad (6)$$

More details about our method are provided in Appendix A.2.

## 4 EXPERIMENTAL RESULTS

In this section, we conduct experiments evaluating our proposed method, MaskCount. We first introduce the evaluation metrics and implementation details, then compare our method with several state-of-the-art methods across different datasets. Finally, we conduct ablation studies to assess the impact of our key designs.

### 4.1 METRICS

We evaluate performance using two commonly used regression metrics: Mean Absolute Error (MAE) and Root Mean Squared Error (RMSE), which quantify the difference between predicted and ground truth values. MAE reflects the estimation accuracy, while RMSE captures its stability.

### 4.2 IMPLEMENTATION DETAILS

All experiments are conducted on 4 NVIDIA H100 GPUs. Performance is evaluated by calculating MAE and RMSE between the model predictions and ground truth values.

In the first stage, we use the pre-trained CLIP Radford et al. (2021) model with ViT-B/16 Dosovitskiy (2020) as the backbone. The backbone parameters are frozen, while all other parameters are trained on DIOCblueberry training set. We train for 200 epochs using InfoNCE loss, with a batch size of 128 and the AdamW optimizer with a learning rate of $1 \times 10^{-4}$. The entire training process takes approximately 3 hours on 4 NVIDIA H100 GPUs.

In the second stage, the model utilizes the SwAV pre-trained ResNet50 He et al. (2016) as the backbone. The backbone network parameters are frozen. All other parameters are trained for 60 epochs using the AdamW optimizer, with a learning rate of $1 \times 10^{-4}$ and a weight decay of $1 \times 10^{-4}$. The contrastive loss weight in Eq. (6) is set to $\lambda_{con} = 1 \times e^{-2}$. We train for approximately 1.5 hours on 4 NVIDIA H100 GPUs with a batch size of 2.

### 4.3 COMPARISON WITH STATE-OF-THE-ART METHODS

Table 2 presents a comparison of our method with several state-of-the-art methods on DIOCblueberry and two general counting datasets. As shown in the table, our method outperforms previous counting methods on the challenging DIOCblueberry dataset and achieves state-of-the-art performance on both almond dataset and tomato detection dataset. Additionally, for high-resolution inputs (avg. $3391 \times 3771$), our inference speed is 10.95 FPS on an RTX 3090Ti.

Due to the small size of blueberry fruits, the two-stage method CounTR Liu et al. (2022) struggles to count objects effectively on DIOCblueberry. In contrast, our two-stage multi-modal method, MaskCount, drastically improves performance, reducing MAE from 491.35 to 38.34 and RMSE from 993.91 to 55.32. When compared to the multi-modal method CLIP-Count Jiang et al. (2023), MaskCount achieves a notable reduction in MAE by 54.85% and RMSE by 61.66%. Additionally, MaskCount surpasses ViT-based CACVIT Wang et al. (2024) and ResNet-based models: LOCA ukić et al. (2023), SAFECount You et al. (2023), and SSD Xu et al. (2024).

Table 2: Performance comparison between our method and state-of-the-art methods on different datasets

| Method | Year | almond dataset | | tomato detection dataset | | **DIOCblueberry (ours)** | |
|---|---|---|---|---|---|---|---|
| | | MAE | RMSE | MAE | RMSE | MAE | RMSE |
| CounTR | 2022 | 5.26 | 6.86 | 4.97 | 6.48 | 491.35 | 993.91 |
| LOCA | 2022 | 2.56 | 3.42 | 2.34 | 3.18 | 51.21 | 85.33 |
| SAFECount | 2022 | 2.66 | 4.01 | 2.63 | 3.75 | 59.78 | 109.61 |
| CLIP-Count | 2023 | 5.63 | 6.87 | 7.53 | 9.15 | 84.91 | 144.29 |
| CACVIT | 2023 | 5.76 | 7.03 | 5.23 | 6.87 | 72.32 | 121.29 |
| SSD | 2024 | 3.95 | 5.57 | 3.53 | 4.96 | 106.01 | 270.00 |
| **ours** | 2025 | **2.03** | **3.02** | **1.91** | **2.71** | **38.34** | **55.32** |

We also conduct experiments of the crowd counting method P2PNet and the indiscernible object counting method IOCFormer on DIOCblueberry. Table 3 compares the counting performance of our method with IOCFormer and P2PNet on DIOCblueberry. IOCFormer represents the state-of-the-art on IOCfish5K—the largest existing dataset for indiscernible object counting. While P2PNet is a representative crowd counting method. Despite the strong baselines, our method con-

Table 3: Comparison of our method with crowd counting and indiscernible object counting methods

| Method | DIOCblueberry | |
|---|---|---|
| | MAE | RMSE |
| P2PNet | 53.23 | 75.01 |
| IOCFormer | 66.86 | 107.63 |
| MaskCount(ours) | **38.34** | **55.32** |

sistently achieves the best performance, further highlighting its robustness and effectiveness. These results also underscore the inherent difficulty of our agricultural DIOC scenario, which poses significant challenges beyond those in existing datasets.

## 4.4 ABLATION STUDY

We conduct a comprehensive ablation study to illustrate the contributions of our design components: CLIP-based mask generation, the edge-aware patch cropping mechanism, and the contrastive loss.

The results presented in Table 4 demonstrate that each component of our design contributes to performance improvement, confirming the effectiveness of every design. Specifically, *Mask* leads to performance improvements, with MAE and RMSE decreasing by approximately 6.33% and 11.56%, respectively.

Table 4: Ablation experiments with different combinations of our key designs. VM: vanilla model. *Mask*: CLIP-based mask generation. *Crop*: our edge-aware patch cropping mechanism. *Con*: our contrastive loss

| Model | DIOCblueberry | |
|---|---|---|
| | MAE | RMSE |
| VM | 51.21 | 85.33 |
| VM+*Mask* | 47.97 | 75.47 |
| VM+*Mask*+*Crop* | 40.81 | 58.27 |
| VM+*Mask*+*Crop*+*Con* | **38.34** | **55.32** |

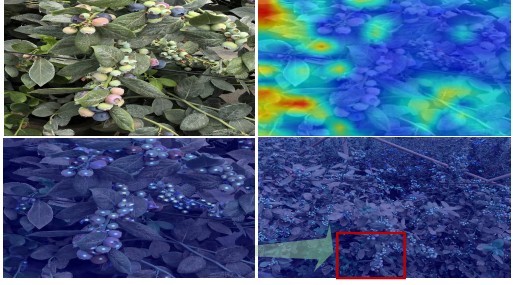

Figure 6: Visualization results of our CLIP-based mask generation to segment foreground and background. *Top left*: original image, *top right*: mask, *bottom*: predicted density map.

We compare the effects of using CLIP and SLIP Mu et al. (2022) for object and background segmentation on counting performance. As shown in Table 5, the model achieves optimal counting performance when using the pre-trained CLIP model with ViT-B/16 Dosovitskiy (2020) as the backbone.

Figure 6 presents the visual results of our CLIP-based mask generation. In addition, we compare the counting performance under different prompts in Appendix A.3.

Figure 7: Final counting results visualization of general cropping compared to ours. *Left*: general cropping method, *right*: our edge-aware patch cropping mechanism. The red circle marks a blueberry located in the edges of the cropped image patches.

Table 5: Comparison of different backbones

| Model | Backbone | DIOCblueberry | |
|---|---|---|---|
| | | MAE | RMSE |
| SLIP | ViT-B/16 | 41.78 | 60.55 |
| CLIP | ViT-B/32 | 46.62 | 62.90 |
| CLIP | ViT-B/16 | **38.34** | **55.32** |

In Table 6, our edge-aware patch cropping mechanism improves performance, reducing MAE and RMSE by approximately 6.10% and 14.44%, respectively. Furthermore, our edge-aware patch cropping mechanism outperforms the general cropping method by reducing MAE from 41.81 to 38.34 and RMSE from 58.80 to 55.32. Details on the selection of padding size for our crop are provided in Appendix A.3.

As shown in Figure 7, the image on the *left* shows the counting results using general cropping. The blueberry in the red circle is located at the edge of the cropped image patch and is counted twice. On the *right*, the counting results using our edge-aware patch cropping mechanism are shown, where the same blueberry is counted only once. These results demonstrate that our method effectively eliminates stitching artifacts by disregarding edge predictions.

As shown in Table 7, we compare the performance of different contrastive losses. Our contrastive loss leads to a reduction in MAE by approximately 6.05% and RMSE by nearly 5.06%.

Table 6: Comparison of different cropping methods

| Method | DIOCblueberry | |
|---|---|---|
| | MAE | RMSE |
| Resize | 40.83 | 64.66 |
| General crop | 41.81 | 58.80 |
| Our crop | **38.34** | **55.32** |

Table 7: Comparison of different contrastive losses

| Loss | DIOCblueberry | |
|---|---|---|
| | MAE | RMSE |
| No Loss | 40.81 | 58.27 |
| InfoNCE Loss | 41.64 | 60.04 |
| Our contrastive loss | **38.34** | **55.32** |

More visual results are provided in Appendix A.4. In addition, the analysis of our limitations is provided in Appendix A.5.

## 5 CONCLUSION

In this paper, we introduce a novel task called dense and indiscernible object counting (DIOC), which presents significant challenges due to the small size, dense distribution, and visual ambiguity of the objects. To facilitate research on DIOC task, we use blueberries as our initial target. We introduce a specialized dataset, DIOCblueberry, which surpasses any general counting dataset in complexity. To address these challenges, we propose MaskCount, a two-stage multi-modal method. In the first stage, MaskCount segments objects from complex backgrounds to simplify the images for counting. In the second stage, we apply a novel contrastive loss to enhance the separation between the objects and the background. Given the high resolution of real-world images, we propose an edge-aware patch cropping mechanism that generates overlapping patches to improve counting accuracy and mitigate edge artifacts caused by traditional cropping methods. Extensive experiments demonstrate the superiority of proposed method MaskCount and the effectiveness of our designs. In future work, we plan to expand our DIOCblueberry to other DIOC scenarios and further explore DIOC task.

## 6 ETHICS STATEMENT

This work focuses on a novel task termed dense and indiscernible object counting (DIOC), for which we construct a specialized dataset, DIOCblueberry, and design a method to achieve better counting accuracy. To facilitate research on DIOC, we introduce the DIOCblueberry dataset. The dataset was collected and constructed by the authors. To address its challenges, we propose MaskCount, a two-stage multi-modal approach. Upon acceptance, we will release the full package—including all code, datasets, evaluation benchmarks, and model checkpoints—under the CC BY 4.0 license to ensure maximum reusability. This practice adheres to the principles of open science while maintaining proper attribution and respect for licensing terms. Our work does not involve human subjects or sensitive information; both the models and datasets are solely intended to advance education and scientific discovery.

## 7 REPRODUCIBILITY STATEMENT

We have undertaken several measures to ensure the reproducibility of our work. The details of the model architecture and evaluation protocols are provided in Section 3 and Section 4, while additional implementation details, training configurations, and hyperparameter settings are included in Appendix A. Our dataset was collected and constructed by the authors, with the data acquisition and dataset construction procedures described in Section 2. Upon acceptance, we will release all code, datasets, evaluation benchmarks, and model checkpoints under the CC BY 4.0 license to maximize transparency and reusability.

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

# A  APPENDIX

## A.1  DATASET

### A.1.1  DIOCBLUEBERRY

Figure 8 illustrates the proportion of images within each object count range across different datasets. The results indicate that the majority of images in general counting datasets contain fewer than 100 objects. In contrast, a significant proportion of images in our DIOCblueberry dataset contain more than 100 objects, and some even more than 1000.

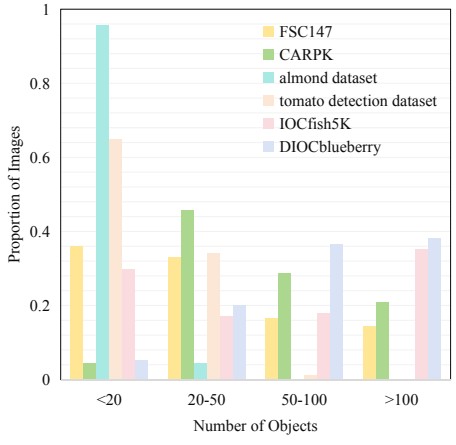

Figure 8: Histogram of images distribution across various count ranges.

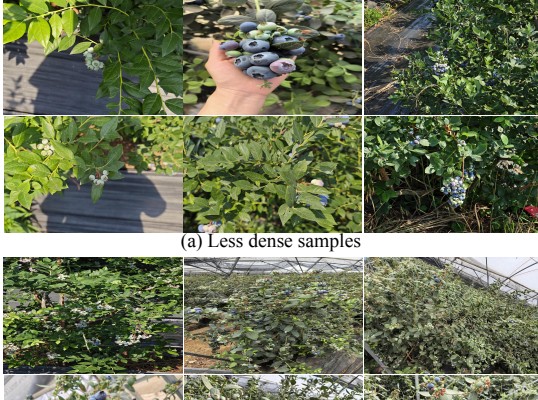

(a) Less dense samples

(b) Dense samples

Figure 9: More examples of our DIOCBlueberry.

Figure 9 shows more examples of DIOCblueberry, with *(a)* depicting less dense images and *(b)* depicting dense images. From left to right in *(a)*, the object count ranges are ≤20, 20–50, and 50–100. While in *(b)*, the object count ranges are 100–500, 500–1000, and >1000. This demonstrates that our DIOCblueberry includes a diverse set of images, covering varying densities of objects from sparse to dense. It effectively showcases the diversity in object distribution and is capable of handling counting tasks across different object densities. This demonstrates the generalization ability of our method.

### A.1.2  COMPARISON WITH OTHER DATASETS

Figure 10 presents a visual comparison between DIOCblueberry and general counting datasets. Compared to all these datasets, our scenes are significantly more complex, featuring indiscernible objects and higher object densities. These characteristics make the counting task in DIOC considerably more challenging and representative of real-world deployment scenarios.

Figure 11 provides a visual comparison between DIOCblueberry and IOCfish5K, the largest indiscernible object counting dataset. While IOCfish5K contains visually ambiguous objects, DIOCblueberry introduces additional challenges. The objects exhibit high visual similarity to the background, scenes are significantly more cluttered, and occlusions are more severe. These factors collectively make DIOCblueberry a more complex and demanding benchmark for evaluating counting performance in real-world scenarios.

As shown especially in the bottom row, although IOCfish5K also contains a large number of objects, they are densely clustered in localized regions. In contrast, our objects are more widely dispersed across the scene, making the counting task in our scenes more challenging.

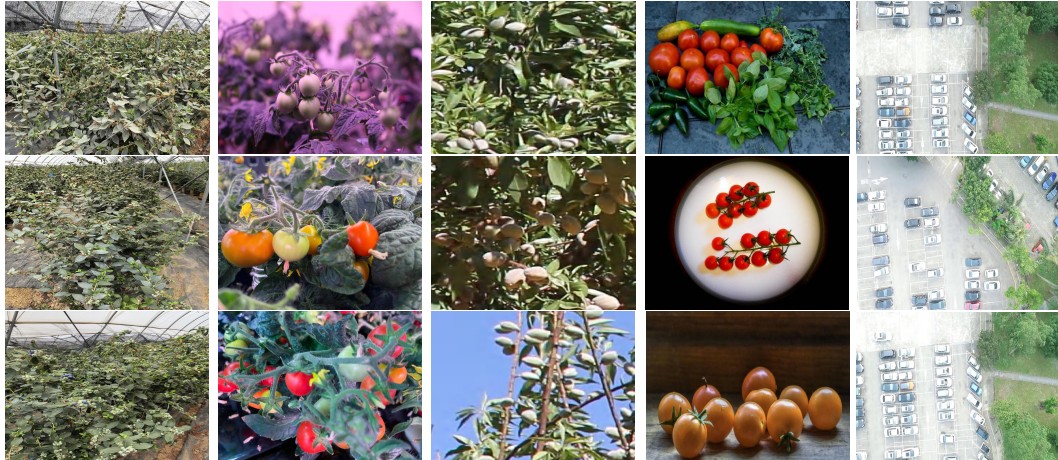

Figure 10: Visual comparison between DIOCblueberry and general counting datasets. *The first column:* our DIOCBlueberry, *the second column:* tomato detection dataset, *the third column:* almond dataset, *the fourth column:* FSC147, *the fifth column:* CARPK.

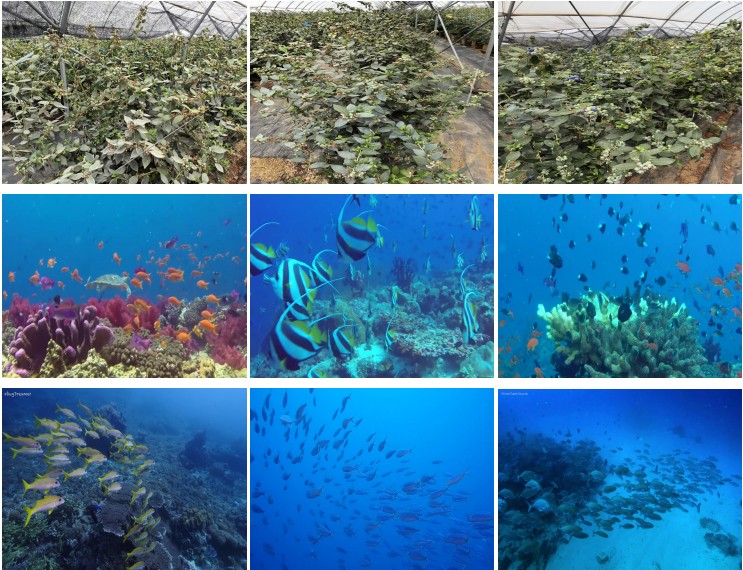

Figure 11: Visual comparison between DIOCblueberry and IOCfish5K. *Top:* our DIOCBlueberry, *middle and bottom:* IOCfish5K.

## A.2 IMPLEMENTATION DETAILS

As our DIOCBlueberry employs point and box annotations, we should transform these discrete points into continuous density maps. For density map generation, images and point annotations are first rescaled to the target resolution, and a sparse impulse map is initialized at the rescaled point locations. Gaussian filtering is then applied to diffuse the discrete points into a continuous distribution: when reference boxes are available, the kernel size is adaptively determined by the average box scale; otherwise, a fixed kernel is used. This process preserves the spatial distribution of annotations while producing smooth and scale-consistent density representations that serve as continuous supervisory signals for model training.

In our implementation, we employ the large vision-language model Qwen2.5-VL-72B to generate a list of background descriptive words, with the input prompt specified as "Please summarize the background area of this image that excludes all regions related to the specified target class blueberry, covering as much non-class area as possible. Use a list of single words (within 10 words, primarily

nouns) and output only the words list, with different words separated by commas, without any additional text".

For the contrastive loss $\mathcal{L}_{stage1}$ 1, taking the derivative with respect to the similarity score $s_k = \text{sim}(\varepsilon_p^k, \varepsilon_t)/\tau$ yields: for a positive sample $k$, $\frac{\partial \mathcal{L}}{\partial s_k} = -p_k^{\text{pos}} + p_k^{\text{all}}$, where $p_k^{\text{pos}} = \frac{e^{s_k}}{\sum_{i \in \mathcal{P}} e^{s_i}}$ and $p_k^{\text{all}} = \frac{e^{s_k}}{\sum_{j \in \mathcal{P} \cup \mathcal{N}} e^{s_j}}$. For a negative sample $k$, $\frac{\partial \mathcal{L}}{\partial s_k} = p_k^{\text{all}}$.

This indicates that the loss encourages the entire positive set to dominate the softmax distribution, with the intra-positive distribution serving as the ideal target. Compared to the conventional single-positive InfoNCE, this formulation distributes gradients across multiple positives, reducing training variance and instability. Additionally, it robustly aggregates intra-class diversity and mitigates the effect of false negatives. And it imposes consistency constraints over the whole positive set, thereby learning more compact and discriminative feature representations.

## A.3 ABLATION STUDY

Table 8 shows the counting results corresponding to CLIP-based mask generation using different prompts in first stage. We conduct experiment using class name as prompt to identify foreground objects for counting. However, results show that this approach underperforms using background-descriptive prompt to mask out irrelevant regions before counting. This highlights the effectiveness of background suppression over foreground guidance in DIOC scenes.

Table 8: Ablation experiments with different background prompts of our CLIP in the first stage. *Top 1*: the most similar text selected from the candidate texts, *top 3*: top three most similar texts selected from the the candidate texts

| Prompt type | Prompt | DIOCblueberry | |
| --- | --- | --- | --- |
| | | MAE | RMSE |
| foreground | "blueberry" | 47.49 | 91.18 |
| | "leaf" | 39.74 | 59.52 |
| background | *top1* (ours) | **38.34** | **55.32** |
| | *top3* | 41.51 | 58.44 |

Table 9 shows the results of our edge-aware patch cropping mechanism with different padding sizes, indicating that a padding size of 32 leads to a lower counting MAE and RMSE. This suggests that using a padding size of 32 effectively minimizes the interference from edge effects, improving the accuracy of bounding box positioning. As a result, our model is able to make more accurate predictions of object counts.

The experiments on background prompt selection demonstrate that using the highest-ranked prompt performs better than using just "leaf" or the top three prompts. These results indicate that selecting the most relevant background prompt for mask generation can significantly improve counting accuracy. By choosing the most relevant prompt, we can more effectively separate the objects from the background, thereby enhancing the overall performance of DIOC task.

## A.4 SUBJECTIVE PERFORMANCES

Our CLIP-based mask generation alleviates the counting challenges posed by complex backgrounds. It segments the objects and background, which improves the quality of the final density maps and leads to better object counting performance. The heatmaps and the segmentation results of our CLIP-based mask generation are shown in Figure 12, in the heatmaps, the background areas are activated. In the visualization images, the background areas are masked. In the first four rows, we visualize the mask results for four different object densities in DIOCblueberry, showing that our CLIP-based mask generation can accurately segment the objects and background, whether the objects are

Table 9: Ablation experiments with different padding sizes of our edge-aware patch cropping mechanism on DIOCblueberry

| Padding size | DIOCblueberry | |
| --- | --- | --- |
| | MAE | RMSE |
| 16 | 45.26 | 65.56 |
| 32 (ours) | **38.34** | **55.32** |
| 64 | 44.09 | 69.75 |

sparse or dense. As shown in the last two rows in Figure 12, we display the visualized counting results for two general counting datasets: the almond dataset and tomato detection dataset. The results demonstrates that our CLIP-based mask generation performs effectively on general counting datasets as well. Our CLIP-based mask generation effectively segments the objects and background, reducing the background complexity and lowering the difficulty of the subsequent object counting task.

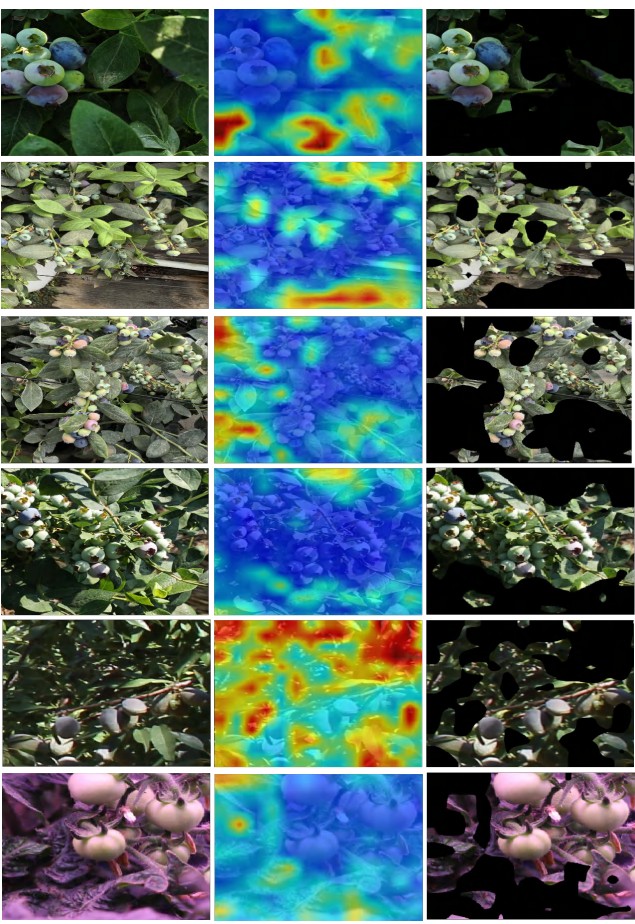

Figure 12: Visualization results of our CLIP-based mask generation to segment foreground and background. *The fist four rows*: our DIOCBlueberry, *the fifth row*: the almond datast, *the last row*: tomato detection dataset.

The results of CounTR Liu et al. (2022), LOCA ukić et al. (2023), SAFECount You et al. (2023), CLIP-Count Jiang et al. (2023), CACVITWang et al. (2024), SSD Xu et al. (2024) and ours on our DIOCBlueberry, almond dataset, and tomato detection dataset are shown in Figure 13. In our DIOCblueberry, the object count per image ranges from 10 to 2000. In the first four columns, we visualize the counting results of four different images with ground truth counts ranging from 10 to around 2000. The results demonstrate that our method, MaskCount, can handle both sparse and dense counting scenarios. As shown in the last two rows in Figure 13, our method is able to count the objects that are small in size, dense distribution, and visual ambiguity with their surroundings in complex backgrounds. It demonstrates superior performance on DIOCBlueberry compared to state-of-the-art methods. The last two columns of Figure 13 show the visualized counting results on general counting datasets: the almond dataset and tomato detection dataset. These results show that our MaskCount also performs excellently on general counting datasets, demonstrating the versatility of our method.

## A.5 LIMITATION ANALYSIS

While MaskCount achieves significant improvements in counting accuracy, its inference speed and computational cost are higher compared to one-stage methods such as LOCA and SSD. This suggests a trade-off between accuracy and efficiency. However, this limitation is not fundamental and can be addressed. As a promising direction for future work, we plan to distill MaskCount into a lightweight, end-to-end model to better meet the demands of real-world deployment scenarios.

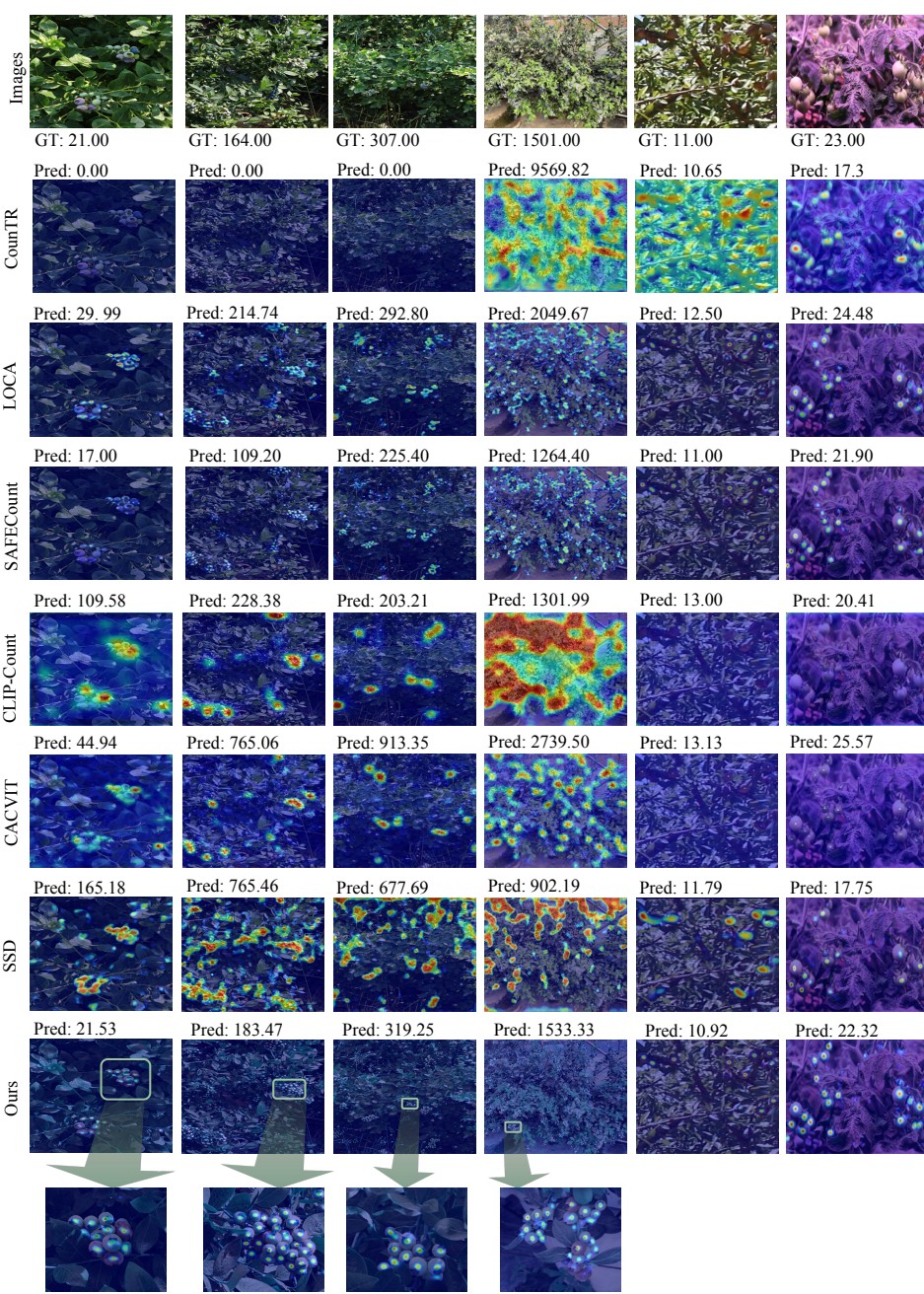

Figure 13: Qualitative results on our DIOCBlueberry, almond dataset, and tomato detection dataset. *The first four columns:* our DIOCBlueberry, *the fifth column:* almond dataset, *the sixth column:* tomato detection dataset (bottom).

