# OpenReview forum: "Dense and Indiscernible Object Counting in Agricultural Scenes"
_ICLR.cc/2026/Conference — ICLR 2026 Conference Withdrawn Submission_

### Official Review · Reviewer_x6jy · 2025-10-21

**Soundness:** 2
**Presentation:** 2
**Contribution:** 2
**Rating:** 2
**Confidence:** 5

**Summary:**

The paper introduces the task of dense and indiscernible object counting (DIOC), which involves counting objects that are typically small in size and exhibit strong visual ambiguity with surroundings. To facilitate this task, a new dataset dubbed DIOCblueberry is proposed, comprising 6,265 in-field blueberry images annotated with a total of 679,030 objects. To address the challenges of DIOC, the authors propose a two-stage multi-modal counting method that follows a segment-then-count paradigm, featuring an edge-aware cropping mechanism designed for high-resolution images. Experimental results show that the proposed method performs favorably against previous counting methods on the DIOCblueberry dataset.

**Strengths:**

* The paper introduces dense and indiscernible object counting, a challenging counting task with applications such as crop yield estimation and fruit harvest planning.
* To support research on DIOC in agricultural scenes, the DIOCblueberry dataset is presented, containing 6,265 high-resolution blueberry images captured in real-world field environments.
* A two-stage multi-modal method is proposed for DIOC, achieving state-of-the-art results on the proposed dataset.

**Weaknesses:**

* The idea of segment-then-count is not new in object counting. Several prior works [1,2,3] have already leveraged segmentation to enhance counting accuracy.
* Complex pipeline. The proposed pipeline involves multiple steps, including cropping, image mask generation (Stage 1), density map prediction (Stage 2), and stitching. This multi-step design would increase inference time and computational overhead.
* The necessity of the proposed pipeline requires further justification. As shown in Table 3, P2PNet already surpasses most compared methods without using visual exemplars. This implies that a well-designed single-stage crowd counting model may be sufficient for DIOC.
* Following the previous comment, Table 4 indicates that cropping technique accounts for the majority of performance gain. It is unclear whether existing counting methods could achieve comparable results when combined with cropping technique. If so, the necessity of the proposed two-stage pipeline would become questionable.
* Limited test set size. As noted in Section 2.2, the test set only contains 35 images, which is significantly smaller than the training set (3,759 images). This limited test set is insufficient to reliably assess the robustness and generalization capability of the proposed method.
* The proposed dataset contains only a single object class (i.e., blueberry). In this setting, the use of exemplars may be redundant and lacks clear justification. To demonstrate broader applicability beyond a specific fruit type, the dataset should ideally encompass multiple object classes for evaluation, similar to FSC-147. Additionally, the almond and tomato detection datasets contain fewer than 100 objects per image. This low object density falls far short of the "dense and indiscernible" regime that defines the DIOC task. As such, evaluations on these datasets do not sufficiently validate the superiority of the proposed method.
* Missing comparisons with latest multi-modal counting methods, e.g., CountGD [4].
* Missing related work section.

**Reference**

[1] Understanding the Impact of Mistakes on Background Regions in Crowd Counting. WACV 2021.

[2] Regressor-Segmenter Mutual Prompt Learning for Crowd Counting. CVPR 2024.

[3] Learning to Count from Pseudo-Labeled Segmentation. WACV 2025.

[4] CountGD: Multi-Modal Open-World Counting. NeurIPS 2024.

**Questions:**

1. Did the authors train the compared methods on the proposed DIOCblueberry dataset? If not, the comparisons in Table 2 become unfair, as other methods are trained on the FSC-147 dataset. For a fair comparison, the authors should ensure consistent training and testing protocols.
2. What is the testing setting of the compared methods? Did the authors apply image resizing or cropping during inference? Intuitively, input resolution would have a certain impact on the performance.

---

### Official Review · Reviewer_Ejr4 · 2025-10-29

**Soundness:** 3
**Presentation:** 2
**Contribution:** 3
**Rating:** 6
**Confidence:** 5

**Summary:**

The paper introduces the task of Dense and Indiscernible Object Counting (DIOC) in agricultural settings. It focuses on the challenge of counting small, densely packed objects that are visually indistinguishable from each other. To facilitate research in this area, the paper presents the DIOCblueberry dataset, a new collection of 6,265 high-resolution images with over 679,000 point annotations of blueberries at various growth stages from two farms in China. To address the task of estimating accurate counts from these images, the paper proposes a two-stage, multi-modal counting method called MaskCount.

Stage 1 uses a vision-language model (CLIP) to segment the images into foreground/background. Stage 2 then estimates a density map from the masked images, using a contrastive loss to enhance feature separation. Experiments show that MaskCount significantly outperforms several state-of-the-art methods on the DIOCblueberry dataset as well as on other datasets.

**Strengths:**

1. The main strength is the DIOCblueberry dataset. It is large and high-resolution, and it captures a challenging real-world counting scenario (dense, small, and camouflaged objects). New public benchmarks are critical for advancing the field, in my opinion.
2. The proposed MaskCount method achieves state-of-the-art results on the new dataset, significantly outperforming six other methods, including those designed for general and indiscernible counting. The method also shows good performance on other agricultural datasets.
3. The two-stage approach, particularly the idea of using a VLM to generate a background mask to simplify the scene for a dedicated counter, is logical and proven effective by the ablation studies.
4. The paper provides comprehensive ablation studies in the main paper and appendix, which validate the contribution of each of the method's main components.

**Weaknesses:**

1. The paper's central claim that the DIOC task is "unexplored"  is incorrect. The field of "Indiscernible Object Counting (IOC)" addresses the same challenges. The paper even cites IOCFormer, which makes the "unexplored" claim all the more baffling and undermines the paper's scholarship. In addition, the related work section is sparse and fails to position the work properly against other agricultural counting datasets or the broader IOC literature. Some papers/surveys in the agricultural domain that also address the counting problem are:
* MinneApple: A Benchmark Dataset for Apple Detection and Segmentation
* A survey of public datasets for computer vision tasks in precision agriculture
* Deep-learning-based counting methods, datasets, and applications in agriculture: A review

2. The number of frames in the train/test split is inconsistent, I think. The paper mentions a total of 6,265 images: 3,759 in the training set and 35 in the test set. What happened to the other images? And how exactly were the train/test sets generated? Thirty-five images for a test set also seems really small. Usually, one follows an 80/20 or 90/10 split for train/test data.

3. The paper never explicitly states how the final scalar count (e.g., "Pred: 913.35" ) is derived from the predicted density map. The standard method is to compute the integral (sum) of the map. More importantly, it is not specified if this same extraction method was applied to the density maps produced by the baseline methods (CounTR, LOCA, SSD, etc.) for a fair comparison. This is a crucial detail for verifying the validity of the results in Table 2.

4. The dataset comparison in Table 1 is not complete, in my opinion. It should not be used as a key argument to claim that the proposed dataset is more complex than any other dataset in the literature. This is a too-strong claim that is hard to prove. Instead, in my opinion, it would suffice to highlight the advantages of the presented dataset without having to be the best or most complex.

5. The paper never explicitly states how the final scalar count (e.g., "Pred: 913.35" ) is derived from the predicted density map. The standard method is to compute the integral (sum) of the map. More importantly, it is not specified if this same extraction method was applied to the density maps produced by the baseline methods (CounTR, LOCA, SSD, etc.) for a fair comparison. This is a crucial detail for verifying the validity of the results in Table 2.

6. The process for generating the ground truth density maps is mentioned in the appendix, stating a Gaussian kernel is used, with the size being adaptive if boxes are available (3 per image ) and "fixed" otherwise. How this "fixed" kernel size is chosen and its impact on the ground truth (and thus the MAE/RMSE metrics) is not discussed.

**Questions:**

1. The "edge-aware patch cropping mechanism"  is presented as a key design, but it appears to be a standard overlap-and-stitch tiling method, which is a common technique for processing high-resolution images, no? Or am I missing something here?

2. How is the final count extracted from the density map? Is it by summing all pixel values? Crucially, was this same extraction procedure applied to the predicted density maps of all baseline methods in Table 2  for a fair comparison?

3. How were the baseline methods trained?

4. Can you please clarify the train/test split? Are the 35 test set images from the same two farms as the training data, or do they come from a different, unseen location or distribution? Without this, it's impossible to assess the generalization capability you are testing.

5. For the ground truth density maps, how was the "fixed kernel" size determined for images without exemplar boxes? How sensitive is the model's performance to this choice?

---

### Official Review · Reviewer_pvMD · 2025-10-30

**Soundness:** 2
**Presentation:** 2
**Contribution:** 2
**Rating:** 2
**Confidence:** 5

**Summary:**

This paper introduces a task termed Dense and Indiscernible Object Counting (DIOC) in agricultural scenes and contributes i) a new real world DIOC dataset (DICOblueberry) and ii) MaskCount, a two stage multimodal counting pipeline that first derives a CLIP guided background mask from text prompts, and then estimates density map with masked image, further enhanced with exemplar based cross attention, contrastive regularization, and an edge aware patch cropping scheme. On the proposed test set, MaskCount reports MAE = 38.34 and RMSE = 55.32, and ablations credit improvements to the mask, cropped image patches, and contrastive components.

**Strengths:**

1. **Reasonable Motivation**: This paper presents a clear motivation and a valuable dataset. The statistics convincingly show that the dataset differs from popular counting datasets.
2. **Good Results**: The proposed MaskCount substantially reduces MAE and RMSE compared to several recent counting baselines on DIOCblueberry, and also reports performance gains on two other agricultural datasets.

**Weaknesses:**

1. **Weak Technical Motivation**: The proposed method seems weakly connected to the proposed task. The manuscript does not show how MaskCount can address dense and indiscernible object counting. Furthermore, it states that SOTA methods struggle on DIOC, but does not explain why they underperform.
2. **Limited Technical Contribution**: Compared with prior “segment then count” pipelines. The overall recipe, foreground/background suppression followed by counting, has been explored in several counting approaches such as GeCo[1]. Furthermore, the claimed edge aware patch cropping method seems to resemble the overlap-tile proposed by U-Net.
3. **Misaligned Experimental Setup**: Experimental setup seems strange and misaligned with the central claim. The core claim is to resolve DIOC in agriculture, yet the main comparison is dominated by class-agnostic counting methods, which focus on zero/few-shot generalization counting. Thus, the results are not convincing to back up the DIOC claim, missing comparisons with category-specific counting approaches (e.g., crowd counting approaches have already dense and indiscernible datasets like ShanghaiTech) and with plant counting methods, as they focus on agricultural scenes.
4. **Insufficient analysis of experimental results**: For instance, Table 7 shows that adding InfoNCE degrades performance, yet no mechanism is offered for this strange and non-intuitive result.

[1] Pelhan, Jer, et al. "A novel unified architecture for low-shot counting by detection and segmentation." Advances in Neural Information Processing Systems 37 (2024): 66260-66282.

**Questions:**

1. Method and experiment specification leave critical details unexplained. For example, in stage 1, how positive/negative patches are constructed from the cosine similarity map (thresholds? top k? hard/soft labels?)
2. Which parameters are trainable beyond the frozen CLIP backbone?
3. In stage 2, how the negative background patches are sampled, and how the match module is designed?
4. In section 4, the reproducibility of baseline training is not documented, as it only gives hyper parameters for MaskCount.
5. What is the training strategy of baselines, or whether they are retrained on DIOC?

---

### Official Review · Reviewer_s7Pq · 2025-11-01

**Soundness:** 3
**Presentation:** 3
**Contribution:** 3
**Rating:** 8
**Confidence:** 5

**Summary:**

his paper introduces a new computer vision task: Dense and Indiscernible Object Counting (DIOC), focusing on scenarios common in agriculture where objects are small, densely packed, and visually ambiguous with their background. To facilitate research in this area, the authors present a new, challenging dataset named DIOCblueberry, which contains over 6,200 high-resolution images with extensive annotations.

**Strengths:**

This paper successfully defines and motivates the DIOC task, a challenging problem with significant practical importance, particularly for smart agriculture (e.g., yield estimation). This work addresses a clear gap in existing object counting research, which has largely focused on more discernible objects.

The introduction of the DIOCblueberry dataset is a major contribution. The authors provide a detailed description of the data collection and annotation process (involving ~1,700 human hours), highlighting its complexity compared to existing benchmarks like FSC147.

**Weaknesses:**

The paper focuses exclusively on counting metrics (MAE and RMSE). However, since the method generates density maps, it implicitly performs localization. Given that some baselines like CLTR[1] and FIDTM[2] are localization-based, including a brief analysis of localization performance (e.g., using point-based F1-score or precision/recall) would provide a more complete picture of the model's capabilities and the dataset's challenges.

The first stage relies on a large vision-language model (Qwen2.5-VL-72B) to generate background prompts. How robust is the model to variations or potential failures in this prompt generation step? While Appendix A.3 provides an ablation on prompt choice, a discussion in the main paper regarding the consistency and potential limitations of this automated prompt engineering would be beneficial.

[1] An End-to-End Transformer Model for Crowd Localization. ECCV 22.
[2] Focal Inverse Distance Transform Maps for Crowd Localization. IEEE TMM.

**Questions:**

see weakness

---

### Note · Authors · 2025-11-14

I have read and agree with the venue's withdrawal policy on behalf of myself and my co-authors.